# Emotional and Subsequent Behavioral Responses After Unethical Pro-Organizational Behavior: A Meta-Analysis Based Systematic Review

**DOI:** 10.3390/bs15091266

**Published:** 2025-09-16

**Authors:** Lemei Zou, Yixiang Wang, Chuanjun Liu

**Affiliations:** 1Department of Sociology and Psychology, School of Public Administration, Sichuan University, No. 24 South Section 1, Yihuan Road, Chengdu 610065, China; 2023225015043@stu.scu.edu.cn (L.Z.); 2024225015016@stu.scu.edu.cn (Y.W.); 2Institute of Psychology, Sichuan University, No. 24 South Section 1, Yihuan Road, Chengdu 610065, China

**Keywords:** Chinese databases, emotional responses, moral cleansing effect, moral disengagement, moral licensing effect, moral slippery slope, unethical pro-organizational behavior

## Abstract

UPB elicits various and heterogeneous subsequent behaviors through positive and negative emotions, a phenomenon that warrants a comprehensive meta-analysis. This study synthesized 34 studies from both English and Chinese databases (49 independent samples, *N* = 83,810), published between 2016 and 2024. The results reveal that positive emotions (e.g., pride, psychological entitlement) trigger the moral licensing effect of rationalizing further unethical conduct and the conscientiousness effect of enhancing organizational identification and promoting positive behaviors. Conversely, negative emotions (e.g., guilt, shame) drive the moral cleansing effect of motivating reparative moral behaviors. Additionally, negative emotions can also lead to the moral slippery slope effect of inducing unethical conduct. Moreover, moral disengagement was identified as a self-regulatory mechanism that permeates this entire process, enabling employees to navigate the moral conflicts arising from UPB. This study uncovers the dual nature of UPB from an emotional perspective.

## 1. Introduction

Since the concept of unethical pro-organizational behavior (UPB) was introduced by [75] ([75]), scholars have explored this phenomenon extensively from diverse theoretical perspectives over the past decade ([51]; [52]; [64]). As defined by [75] ([75]), UPB refers to employees’ actions that violate core societal values, ethical norms, laws, or legitimate behavioral standards, with the intention of benefiting their organization or its members ([75]; [74]). Although UPB may appear advantageous to organizations in the short term, studies show that it can actually have a negative impact on their reputation and public trust ([35]; [88]). Additionally, it can lead to emotional and behavioral problems for the people involved ([30]; [90]; [98]).

Numerous studies have examined the antecedents of unethical pro-organizational behavior, such as organizational identification and positive social exchange ([12]; [79]), yet research exploring its consequences remains limited ([90]; [99]). Some scholars have investigated employees’ emotional and behavioral responses following UPB engagement. Empirical evidence indicates that UPB performers may experience complex emotional reactions, including pride, shame, and guilt ([67]; [88]). For instance, an employee may feel proud for helping the company secure a cooperation deal, but also experience shame for concealing quality issues with the company’s products during the negotiation ([88]). [48] ([48]) further demonstrated that these conflicting emotional states (guilt and pride) would induce state anxiety and intensify employees’ work-to-life conflict.

Additionally, research has investigated the link between UPB and emotional exhaustion (EE) and psychological entitlement (PE) ([34]; [44]; [91]). Although EE and PE are more frequently identified as antecedents of UPB, these complex affective states have a two-way effects, simultaneously facilitating the occurrence of UPB and influencing its subsequent behaviors. For instance, employees with heightened PE are more likely to engage in UPB and may also exhibit more counterproductive behaviors ([54]). This suggests that such psychological states persist throughout the UPB process, both before and after the act.

Although existing research has established linkage between UPB and emotional responses, there are several critical gaps in the current literature. First, there are inconsistencies in the empirical findings across studies. For instance, [11] ([11]) reported significant positive correlations between UPB and guilt, whereas [77] ([77]) found a negative association between employees’ UPB and anticipated guilt. Second, most studies rely on cross-sectional surveys or short-term retrospective designs, employing different measurement tools without rigorous or consistent control conditions ([11]; [44]; [48]; [67]). This has resulted in divergent estimates of the effect sizes linking UPB to emotional outcomes. Finally, there is a significant disconnect in the literature between studies examining the emotional consequences of UPB and those investigating its behavioral outcomes. While emotions such as guilt and pride are theorized to be key mechanisms that drive subsequent behaviors (e.g., moral cleansing vs. moral licensing) ([44]), empirical tests of this integrated emotion-behavior sequence remain scarce and fragmented.

Based on the above reasons, it is necessary to conduct a comprehensive evaluation of the relationship between UPB and these emotional states. Additionally, the behavioral consequences that may be brought about by the emotions triggered by UPB also need to be explored. Therefore, to address these issues, the present study will explore the potential emotional reactions elicited by UPB through a meta-analysis. The study will also evaluate the direction and strength of the association between UPB and subsequent emotional reactions. Then, it will examine the subsequent behavioral patterns exhibited by individuals after UPB has induced their emotional reactions. Specifically, the study aims to synthesize the emotional and behavioral aftereffects of UPB, providing theoretical and empirical support for future research.

## 2. Literature Review

### 2.1. UPB and Its Emotional Responses

#### 2.1.1. UPB and the Positive Emotional Responses

Recent scholars have increasingly examined the emotional states triggered by UPB. Given the paradoxical duality of UPB—combining benevolent pro-organizational motivation with unethical behavioral attributes, this co-occurring incongruence may elicit complex emotional experiences in individuals. Initially, perceiving the pro-organizational nature of UPB could generate positive affective states among employees. [72] ([72]) indicate that pride is a positive moral emotion that arises when individuals attribute successful or positive events to their own abilities or efforts. [48] ([48]) empirically demonstrated that employees experience pride when they perceive UPB as conducive to organizational interests (e.g., enhancing team performance), because they believe they have “helped the organization”. This finding aligns with [67]’s ([67]) observation that employees report pride when UPB facilitates organizational goal attainment, as such behaviors are construed as organizational contributions. Despite these findings, however, the connection between the two is not always stable. Some research results indicate that there is no significant correlation between UPB and pride ([44]). This inconsistent relationship has aroused scholars’ interest, prompting increasing attention to be paid to the relationship between UPB and emotional reactions, and leading us to consider why employees have varying emotional responses following UPB.

The reason why employees feel proud of UPB may derive from two mechanisms. First, a high degree of organizational identification can reduce the moral sensitivity of employees and blur their perception of the immorality of UPB ([67]; [95]). When employees have a high degree of organizational identification, they may consider their UPB as reflecting organizational loyalty and responsibility, thereby generating a sense of pride, such as “I have contributed to the organization” or “I have sacrificed my personal morality for the organization” ([44]; [47]). Furthermore, the ethical climate within the organization can influence employees’ emotional responses to UPB. According to affective appraisal theory, individual’s cognitive appraisals of events or behaviors can affect their emotional reactions ([39]). The ethical climate within an organization can guide employees’ moral cognitive appraisals of ethical conduct ([49]; [76]), thereby influencing their emotional responses. Based on this theoretical foundation, [88] ([88]) found that the ethical climate moderates post-UPB affective responses; under low ethical climate conditions, UPB elicits stronger pride, whereas under high ethical climate conditions, UPB evokes heightened shame.

Beyond the positive emotion of pride, the pro-organizational attribute of UPB can trigger the positive belief of psychological entitlement. Psychological entitlement refers to an individual’s stable and pervasive belief that they deserve preferential treatment and exemption from social responsibilities ([8]). [63] ([63]) found a significant positive correlation between psychological entitlement and unethical pro-organizational behavior, particularly when employees’ personal goals align with organizational objectives, making them more willing to engage in unethical conduct for organizational benefit. This phenomenon occurs because employees with heightened psychological entitlement typically believe they deserve special treatment within the organization. These beliefs motivate them to rapidly achieve organizational goals through UPB, thereby demonstrating their personal worth or gaining personal benefits. For instance, highly entitled employees may consider lying for organizational interests to be acceptable, as this would position them as organizational heroes ([40]). Furthermore, highly entitled employees may even commit UPB in retaliation against aggressive customers ([64]). Overall, psychological entitlement weakens employees’ internal moral constraints. Those with heightened entitlement tend to rationalize UPB, alleviating guilt while strengthening their willingness to engage in such conduct ([34]; [98]). This suggests that employees’ pro-organizational motives or the outcomes of their pro-organizational behaviors may paradoxically provide a psychological exemption for their unethical actions. However, considering the contradictory results of previous studies on UPB and its positive emotional responses, as well as the lack of stable estimates for the effect size of their relationship, this study intends to conduct a meta-analysis to comprehensively estimate the strength of the relationship between UPB and pride, as well as between UPB and psychological entitlement.

#### 2.1.2. UPB and the Negative Emotional Responses

Negative emotional responses to UPB are mostly triggered by its unethical nature. [80] ([80]) found that individuals reported feeling more guilt after conducting UPB. Similarly, multiple studies have also found significant positive correlations between UPB and guilt ([36]; [44]; [67]). This association likely occurs because UPB inherently involves deceptive practices (e.g., dishonesty and concealment). Such behaviors violate individuals’ internalized moral standards while simultaneously harming stakeholders’ interests. Therefore, employees experience a strong sense of guilt due to moral pressure ([11]; [99]).

In addition to guilt, employees may experience shame when engaging in unethical pro-organizational behavior ([10]; [36]; [88]). Notably, although guilt and shame share conceptual similarities, guilt is more directed at the specific behavior, meaning that an individual feels guilty for immoral actions that have harmed others. Meanwhile, shame is more directed at the individual themselves, meaning that an individual negates their self-worth due to unethical actions ([69]). [88] ([88]) pointed out that implementing unethical pro-organizational behavior can undermine an individual’s moral self-identity, thereby eliciting negative self-evaluations of both conduct and self-image and inducing shame.

Furthermore, anxiety is also closely related to unethical pro-organizational behavior ([41]; [48]; [86]). [23] ([23]) defined anxiety as a negative emotional state triggered by uncertainty and the anticipation of potential threats. Its core feature is hypervigilance towards future adverse outcomes. There are multiple mechanisms for the connection between anxiety and UPB. On the one hand, UPB simultaneously violates moral standards (eliciting guilt) while benefiting organizational interests (generating pride). These contradictory emotions form a strong psychological conflict in employees’ minds, leading them to repeatedly consider the consequences of their actions and continuously worry about the potential risks of engaging in UPB (such as being punished or exposed), thereby resulting in anxiety ([48]). On the other hand, high work stress and performance demands can also cause employees to experience high levels of anxiety ([20]). Within such anxious states, employees have a strong sense of job insecurity, and regard UPB as a coping strategy, which increases the likelihood of them engaging in UPB ([41]; [87]; [96]).

When the cumulative negative emotions resulting from UPB reach a critical level, employees are prone to developing emotional exhaustion, a composite negative affective state ([99]). According to conservation of resources (COR) theory, individuals strive to acquire, protect, and maintain their resources (e.g., emotional, cognitive, and temporal resources). However, when these resources are persistently depleted without adequate replenishment, individuals may experience stress and psychological threat ([27]; [28]). Emotional exhaustion represents the cumulative depletion of an individual’s emotional resources. Given the contradictory nature between behavioral intentions and unethical properties in UPB, employees must expend substantial emotional resources to engage in moral rationalization or regulate cognitive dissonance arising from UPB. This regulatory process may lead to the excessive consumption of emotional resources, ultimately precipitating emotional exhaustion ([89]). [38] ([38]) also identified a significant positive correlation between unethical pro-organizational behavior and emotional exhaustion. Moreover, when there is a lack of support and care within an organization, employees are unable to effectively replenish their depleted psychological resources. This leads to the persistent exacerbation of emotional exhaustion ([38]; [91]). [30] ([30]) further demonstrated that leaders’ UPB similarly triggers emotional exhaustion among employees. [46] ([46]) built on this research, revealing that employees under high-intensity time pressure and task demands are more susceptible to emotional exhaustion, subsequently increasing their likelihood of engaging in UPB. However, other studies have identified a significant negative correlation between emotional exhaustion and UPB in contexts involving unethical leadership and workplace bullying ([55]; [91]). In light of these conflicting results, a comprehensive meta-analytic evaluation of the relationship between UPB and emotional exhaustion is necessary.

### 2.2. Emotional Responses of UPB and Subsequent Behaviors

#### 2.2.1. Moral Cleansing Effect

Unethical pro-organizational behavior evokes different emotional responses in employees, with these distinct emotional states subsequently influencing their post-UPB conduct. Empirical evidence shows that employees are more likely to offer constructive suggestions to compensate for ethical transgressions if they feel guilty about implementing UPB ([80]). Similarly, [98] ([98]) pointed out that post-UPB guilt reinforced organizational citizenship behaviors while reducing self-interested unethical conduct. [11] ([11]) further identified guilt as a mediating mechanism between UPB and moral behaviors. On the one hand, guilt enhances employees’ customer service behavior, and on the other, it inhibits their self-serving cheating. Overall, previous studies consistently demonstrate that, after engaging in UPB, employees’ guilt fosters their prosocial behavioral tendencies ([44]; [67]).

The increased engagement in prosocial behaviors by employees, following guilt triggered by UPB, may be attributed to a moral cleansing effect. The moral cleansing effect refers to the psychological mechanism through which individuals alleviate guilt and restore their moral self-image by engaging in moral actions that compensate for unethical behaviors ([80]; [98]). This phenomenon constitutes a manifestation of moral self-regulation, whereby the unethical nature of UPB threatens individuals’ moral self-concept. The guilt resulting from such unethical conduct then motivates compensatory moral behaviors ([71]; [99]). For instance, an employee who concealed product defects from customers might proactively suggest ways to improve the product to the company. This behavioral adjustment serves to alleviate the guilt triggered by the initial unethical act and restore moral equilibrium.

Similar to the moral cleansing effect, the moral compensation effect is another phenomenon in which more moral behaviors emerge following unethical behaviors. Scholars first identified this phenomenon decades ago. [101] ([101]) experimentally demonstrated that participants who were asked to imagine making unethical decisions subsequently exhibited heightened moral behavior in follow-up tasks. Although moral cleansing and moral compensation share conceptual overlap (both involving increased ethical behavioral tendencies following unethical acts), moral cleansing specifically emphasizes the motivational role of negative moral emotions, such as guilt. In contrast, moral compensation encompasses a broader range of behaviors aimed at restoring moral psychological equilibrium ([98]; [101]).

In terms of the target of the moral action, moral behaviors stemming from the moral cleansing effect are more focused on regulating the actor’s own negative emotions. Their core purpose is to eliminate the guilt resulting from unethical behaviors, representing a form of moral equilibrium at an emotional level ([99]). For example, leaders who abuse their power may exhibit more caring behaviors to get rid of their feelings of guilt ([43]). As to the moral compensation effect, the actor’s subsequent moral behaviors are more driven by cognitive strategies ([101]). Regardless of whether the actor feels guilty about their own unethical behaviors, they seek to repair their damaged moral image through overt moral behaviors (such as volunteering, donating, ethical voice, etc.) ([85]). The core lies in compensating for the moral deficit caused by unethical behaviors, thereby making oneself “seem moral”, representing a form of cognitive moral equilibrium ([22]). Furthermore, [83] ([83]) noted that both restitution cleansing and prosocial “moral crediting” behaviors are categorized as types of moral cleansing. Consequently, the present study defines moral cleansing as, post-UPB, involving either enhanced moral conduct or reduced unethical behavior, provided that these behavioral changes are mediated by negative moral emotions (guilt and shame). Given the inconsistencies in prior studies, the present study intends to estimate the overall strength of the moral cleansing effect.

#### 2.2.2. Emotional Rationalizing Effect

In addition to restoring moral equilibrium through increased ethical conduct, individuals engage in moral self-regulation via moral disengagement. Moral disengagement refers to the psychological process by which individuals rationalize unethical behaviors through a set of cognitive mechanisms and strategies ([6]). [41] ([41]) demonstrated in their study that high performance pressure could induce workplace anxiety and unethical pro-organizational behaviors among employees while also elevating moral disengagement levels. [44] ([44]) further found that, post-UPB, pride, guilt, and psychological entitlement all exhibited significant positive correlations with moral disengagement. This suggests that moral disengagement behaviors play a significant role in the psychological mechanism of UPB, and these behaviors are related to both positive and negative emotions.

Other studies have generally reported a significant positive correlation between psychological entitlement and moral disengagement ([40]; [94]; [100]). However, conflicting evidences from [34] ([34]) and [77] ([77]) indicated negative correlations between guilt, psychological entitlement, and moral disengagement. This indicates that whether UPB triggers positive or negative emotions in individuals, employees are likely to exhibit moral disengagement behaviors afterwards. Through these behaviors, employees can justify their unethical behaviors and rationalize their feelings. Therefore, the study conceptualizes these cognitive regulatory behaviors, which demonstrate broad associations with both positive and negative emotions following UPB, as the emotional rationalizing effect.

Unlike the moral compensation and moral cleansing effects, the moral disengagement theory emphasizes the cognitive mechanisms through which employees resolve moral self-regulatory dissonance by reconstructing their cognitive appraisals. Specifically, the core mechanism of emotional rationalizing effect is the moral self-regulatory, which refers to the process through which individuals monitor, evaluate, and adjust their behavior to align it with their internal moral standards via intrinsic cognitive and affective mechanisms ([5]). Employees may employ strategies such as moral justification, displacement of responsibility, and dehumanization to rationalize unethical behaviors ([6]). For instance, employees might justify deceiving customers as necessary for organizational survival, attribute responsibility for UPB to supervisors by claiming coercion, or devalue victims by perceiving clients as unsympathetic or deserving of deception ([42]; [82]; [100]). Consequently, this study categorizes morally disengaged behaviors associated with moral emotions during the UPB process as emotional rationalizing effect. Due to inconsistencies in previous research findings, it is necessary to conduct a comprehensive evaluation of this effect.

#### 2.2.3. Moral Slippery Slope Effect

However, negative emotions elicited by UPB are not always effectively regulated. When self-regulation is insufficient, these negative emotions may lead to subsequent unethical behaviors or may diminish moral conduct. [44] ([44]) demonstrated that UPB triggers guilt in employees, which exhibits a significant positive correlation with workplace deviance; the stronger the guilt experienced, the greater the subsequent deviant behavior. Similarly, [94] ([94]) found that UPB exacerbates employees’ self-control depletion, which in turn promotes counterproductive work behaviors. Anxiety and emotional exhaustion associated with UPB also show significant positive correlations with other unethical behaviors ([55]; [96]). Beyond reinforcing unethical conduct, negative emotions can weaken moral behavior. [34] ([34]) noted a significant negative correlation between guilt and organizational citizenship behavior; the stronger the guilt felt by employees after engaging in UPB, the weaker their propensity towards organizational citizenship behavior.

This phenomenon, whereby negative emotions triggered by UPB elicit more unethical conduct or a weakening of ethical standards, can be conceptualized as a moral slippery slope effect. The moral slippery slope effect refers to the gradual relaxation of moral standards among individuals or groups, leading to an increase in unethical behavior ([100]). While the moral slippery slope effect usually focuses on the linkage between initial unethical acts and subsequent unethical behaviors ([81]; [94]; [100]), this study also categorized the connection between emotions triggered by UPB and subsequent unethical behavior as moral slippery slope effect. This is because [70] ([70]) emphasized that moral behavior elicits corresponding moral emotions, which in turn influence an individual’s subsequent moral performance. As to UPB, the unethical nature of UPB obscures individuals’ perception of moral boundaries. When employees perform UPB under the pretext of benefiting the organization, their moral standards gradually erode ([94]), making them more likely to exhibit self-serving unethical behaviors in the workplace. Consequently, this study attributes the relationship between UPB-induced negative emotions and unethical behavior to the moral slippery slope effect and conducts a comprehensive evaluation of its effect size.

#### 2.2.4. Moral Licensing Effect

The positive affective states evoked by UPB (e.g., pride and psychological entitlement) may impair job performance and encourage subsequent unethical behaviors. [34] ([34]) demonstrated that psychological entitlement arising from UPB significantly diminished employees’ work effort and organizational citizenship behaviors. Empirical evidences further indicate that such psychological entitlement reinforces self-interested unethical behaviors and counterproductive work behaviors ([54]; [98]; [97]). Researchers also found that pride positively predicts workplace deviance ([44]), while psychological entitlement shows significant positive correlations with prosocial rule-breaking, unethical behavior, and workplace deception ([33]; [47]; [100]).

This phenomenon whereby individuals experience positive affective states after engaging in UPB, which subsequently reinforces unethical conduct or diminishes moral behaviors, can be conceptualized as the moral licensing effect. The moral licensing effect refers to the tendency for individuals to perceive themselves as having accumulated moral capital after performing moral or prosocial actions, thereby relaxing their ethical standards for subsequent behaviors and even permitting themselves to engage in unethical conduct ([15]; [98]). When employees commit UPB, the pro-organizational nature of such behavior leads them to perceive themselves as having contributed to the organization, thereby obtaining psychological moral licensing. This psychological license makes employees feel entitled to future unethical actions, as they believe their “moral account” has sufficient moral credit due to prior pro-organizational conduct ([37]; [44]).

The core distinction between moral licensing and moral slippery slope effect is that the licensing effect emphasizes the moral superiority attributed to UPB’s pro-organizational characteristics, thereby exempting individuals from subsequent misconduct. Employees experience positive emotional states in the process. In contrast, the moral slippery slope effect focuses on employees’ adaptive desensitization to unethical behaviors, gradually lowering sensitivity through repeated exposure and ultimately escalating unethical conduct. Employees mainly experience negative emotional states in this process. Therefore, this study classifies the association between positive emotional states following UPB and unethical behavioral tendencies as a manifestation of the moral licensing effect and proposes to conduct comprehensive effect analyses accordingly.

#### 2.2.5. Conscientiousness Effect

Finally, some empirical evidence also suggests that positive emotions following UPB (such as pride and psychological entitlement) may trigger more moral behaviors or enhance work engagement. [88] ([88]) demonstrated that, post-UPB, pride positively predicts employees’ work engagement while negatively predicting job burnout. Researchers have also identified significant positive correlations between psychological entitlement and positive workplace behaviors, including job performance, organizational citizenship behavior, and helping behavior ([40]; [63]; [98]).

The study conceptualizes this phenomenon as the conscientiousness effect, that is, individuals who commit UPB may subsequently develop a heightened sense of organizational responsibility and a greater willingness to contribute, driven by the positive affect arising from UPB. This likely occurs because positive affect induced by UPB leads individuals to overestimate the moral value of their actions while underestimating their detrimental consequences. Such moral purification bias may foster a strong sense of moral identification with the behavior, consequently promoting further positive conduct in workplace. The fundamental distinction between the conscientiousness effect and the moral compensation effect lies in their respective motivational mechanisms. In the moral compensation effect, employees engage in ethical behaviors to restore their moral self-image, constituting an “atonement” for prior unethical conduct. Conversely, in the conscientiousness effect, employees exhibit more positive behaviors because they feel a sense of privilege, making them believe that they have a responsibility to do more, rather than out of guilt or a desire for compensation. Accordingly, this study categorizes the relationship between positive emotional states following UPB and subsequent positive behaviors as the conscientiousness effect and proposes to estimate the overall effect size of this relationship.

In summary, the emotional-behavioral mechanism of unethical pro-organizational behavior can be schematically represented in Figure 1.

## 3. Methods

### 3.1. Literature Search

This study followed the PRISMA protocol for systematic reviews and meta-analysis ([56]). A systematic literature search was conducted on 30 July 2024 across multiple databases, including Web of Science, EBSCO, ScienceDirect, Google Scholar, Scopus, CNKI, VIP, and Wanfang Data. The study searched articles published in the aforementioned databases from 2010 onwards ([75], first introduced the concept of UPB in 2010), using the keywords “unethical pro-organizational behavior” and “pro-organizational unethical behavior” for the English databases, using the keywords “亲组织不道德行为” and “亲组织非伦理行为” (both of these two Chinese keywords refer to unethical pro-organizational behavior) for the Chinese databases. The screening process was conducted by two master’s students majoring in psychology. Both students had systematically studied the knowledge related to meta-analysis, literature searches, and coding, and both possessed the relevant professional expertise. The two individuals independently searched the databases and then combined the search results. The initial search yielded 3573 results, which were reduced to 1180 articles after removing duplicates. Based on the following criteria, (1) publication in SSCI or CSSCI journals; (2) quantitative research design; (3) inclusion of UPB as a core variable, we identified 283 relevant studies through screening titles and abstracts. Further full-text review resulted in the selection of 41 articles focusing on UPB and emotions as core variables. However, a few studies on emotions and UPB were excluded, mainly due to insufficient independent sample sets. For instance, [25] ([25]) explored the relationship between depression and UPB, but the research on their relationship only had this one independent sample set, making it impossible to conduct a comprehensive effect estimation, and thus was excluded. Ultimately, 34 eligible studies were included in the meta-analysis. Details of this process follow the PRISMA protocol as outlined in Figure 2.

### 3.2. Inclusion and Exclusion Criteria

Only studies that met the following criteria were selected for inclusion:

Inclusion criteria: (1) the publication language was English or Chinese; (2) the article was published in SSCI or CSSCI journals; (3) the article was quantitative research design; (4) the variables contain measures of UPB and feelings, such as guilt, pride, and shame; (5) effect size were reported, or enough data were available to calculate them.

Exclusion criteria: (1) for studies of the same feelings, there are less than three independent datasets; (2) the article was published before 2010; (3) qualitative studies, literature reviews, and meta-analyses were excluded.

After reading the full texts, 34 articles that met the standards were finally adopted. Each article contains 1 to 4 independent studies, and ultimately 49 independent datasets related to UPB and emotions were included in the meta-analysis (as shown in the Appendix A).

### 3.3. Data Extraction and Coding

In the inclusion articles of UPB and emotional responses, the author extracted the following information: (1) NO., (2) author and publication year; (3) study No.; (4) emotional responses; (5) sample; (6) type of the original effect size; (7) the original effect size value; (8) convert into the effect size value of Fisher’s Zr; (9) standard error. The data were shown in the Appendix A.

In the inclusion articles of emotional responses and behaviors, the author extracted the following information: (1) NO.; (2) author and publication year; (3) study No.; (4) emotional responses; (5) behaviors; (6) type of the behavior; (7) sample; (8) type of the original effect size; (9) the original effect size value; (10) convert into the effect size value of Fisher’s Zr; (11) standard error.

### 3.4. Quality Assessment

Given that most of the studies included in the meta-analysis were cross-sectional studies, the methodological quality of the included studies was evaluated using an 11—item checklist endorsed by the Agency for Healthcare Research and Quality (AHRQ, [60]). For each item, a score of “0” was assigned for responses of “NO” or “UNCLEAR”, while a score of “1” was given for “YES” responses. The quality of the articles was determined as follows: low quality = 0–3; moderate quality = 4–7; high quality = 8–11 ([31]).

### 3.5. Statistical Analysis Procedure

The present study categorized the included studies into two primary sets. The first set examines the relationship between unethical pro-organizational behavior and emotional responses, while the second set investigates subsequent individual behaviors following UPB and these emotional reactions. In the first set, researchers primarily examined the magnitude of relationships between UPB and pride, guilt, shame, anxiety, emotional exhaustion, and psychological entitlement, respectively. For the second set, researchers classified subsequent behaviors into five distinct patterns: moral licensing, moral slippery slope, moral cleansing, conscientiousness effect and emotional rationalizing effect (for behavioral classification criteria, refer to the preceding section). Researchers then separately analyzed the strength of associations between affective states and these behavioral outcomes.

The effect size statistic used in this meta-analysis is the correlation coefficient (*r*). Researchers extracted the correlation coefficient r from the statistical results reported in original studies. If the original study did not report the r value, then we converted the reported effect size indices to r based on available information. However, since variance affects the pooled estimates of effect size r, all correlation coefficients were transformed into Fisher’s Zr values prior to meta-analysis to stabilize variances ([7]). Subsequently, results were converted back to r for interpretation. All data were analyzed using JASP V 0.19.3.0 to merge effect sizes.

## 4. Results

### 4.1. Description of Studies

The meta-analysis included 34 studies with 112 effect sizes (48 for emotional responses and 64 for behaviors). The sample size of the research ranged from 62 to 674, and the participants were mainly employed employees. The quality of the included studies was assessed through AHRQ. The results showed that 4 studies were of low quality, 27 studies were of moderate quality, and 3 studies were of high quality.

### 4.2. Heterogeneity Test

The heterogeneity tests were conducted to assess the variability in study outcomes. In this research, the Q value and the I^2^ statistic are used to assess heterogeneity. The Q value assesses the presence of heterogeneity, while the I^2^ quantifies the proportion of total variance in the effect size attributable to between-study heterogeneity. For I^2^, [26] ([26]) suggested that values approximating 25%, 50%, and 75% may be indicative of low, moderate, and high heterogeneity, respectively. The results of the heterogeneity test are presented in Table 1. It indicates that the Q value for six emotional responses and five behavioral effects were statistically significant, suggesting a substantial heterogeneity across the studies. The Q statistic reached a significant level (*p* < 0.001), and the I^2^ value exceeded 75%, which indicates strong heterogeneity among the studies. Given the high level of heterogeneity across studies and the relatively limited number of studies and sample sizes, the study employed the Paule–Mandel method to estimate the composite effect size ([2], [3]).

### 4.3. Publication Bias Analysis

Publication bias refers to the phenomenon of effect size deviation due to the higher likelihood of studies with statistically significant results being published ([61]). The study used the fail-safe N to test the impact of possible publication bias on the relationships between variables. The results are shown in Table 2. Except for the emotional exhaustion, the fail-safe N of the other groups of relationships far exceeded the critical value (5K+10, K represents the number of independent samples), proving that there was no serious publication bias problem in this study.

### 4.4. Overall Effect Size Estimation

The study employed the Paule–Mandel method to estimate the composite effect sizes across groups, with detailed results presented in Table 3. According to [14] ([14]), values below 0.19 indicate a very weak correlation; 0.20 to 0.39, a weak correlation; 0.40 to 0.59, a moderate correlation; and above 0.60, a strong correlation.

The meta-analytic results revealed that in the composite effect estimation from unethical pro-organizational behavior to affective responses, all measured emotional states except emotional exhaustion demonstrated weak correlations with UPB (*r* = 0.261–0.339), with 95% confidence intervals excluding zero. This suggests statistically significant yet weak associations between UPB and these affective responses. Emotional exhaustion showed no significant correlation with UPB (*z* = −0.451, *p* = 0.652).

Regarding the composite effect estimation from affective responses to subsequent UPB behaviors, all group effect sizes fell within the 0.233–0.328 range, 95% CIs excluding zero, indicating consistent albeit weak manifestations of moral licensing, moral compensation, moral slippery slope, emotional rationalizing effect, and conscientiousness effects following UPB enactment.

## 5. Discussion

### 5.1. Positive Emotions Elicited by UPB and Subsequent Behavioral Mechanisms

The present study identified stable positive correlations between unethical pro-organizational behavior and pride, guilt, shame, anxiety, and psychological entitlement. Regarding positive effects, UPB may elicit both pride and psychological entitlement in employees. This phenomenon likely occurs because employees perceive heightened pro-organizational intentionality in UPB. This perception leads employees to attribute organizational benefits to personal accomplishment, while framing unethical conduct as sacrificial acts for organizational benefits ([48]; [67]). The positive emotions experienced by employees, engendered by their contributions to the organization, foster a sense of pride and privilege. This sense of entitlement, in turn, can lead to a belief that they are above reproach, even in the event of unethical behavior ([98]). Similarly, [65] ([65]) found that when employees engage in pro-organizational behaviors, such as organizational citizenship behaviors, they perceive themselves as having accumulated moral credits due to their “good deeds”. This perception grants them a sense of psychological entitlement, leading employees to believe that they have the right to deviate from moral norms in the future. This psychological cascade, where UPB-induced psychological entitlement generates perceived justification for future misconduct, constitutes the moral licensing effect.

In addition to triggering the moral licensing effect, positive emotions may also engender conscientiousness effects. Specifically, employees experience positive emotions (pride and psychological entitlement) following UPB, which subsequently reinforces their organizational identification and engagement. This phenomenon may be attributable to two factors. First, employees may become cognizant of their contributions to the organization at elevated levels. Second, they may develop a strong perception of being indispensable organizational members. Such positive affective states encourage individuals to make additional organizational contributions (e.g., exhibiting increased innovative behaviors and organizational citizenship behaviors) ([33]; [63]).

### 5.2. Negative Emotions Elicited by UPB and Subsequent Behavioral Mechanisms

Unethical pro-organizational behavior elicits various negative affect states in actors, including guilt, shame, anxiety, and emotional exhaustion. When UPB causes harm to others, employees experience a sense of guilt. When UPB damages the moral self-image of the employees, shame arises ([80]; [88]). Anxiety and emotional exhaustion are more complex manifestations of negative emotions. When UPB elicits both pride and guilt in employees, the conflicting feelings consequently induce anxiety, which makes employees worry about the consequences of UPB ([48]). Furthermore, unique work environments can also influence employees’ anxiety. For instance, high time pressure and job demands are likely to trigger anxiety, and in such cases, employees are more prone to engage in UPB to cope with the stress ([20]; [41]; [87]).

However, the relationship between emotional exhaustion and UPB is relatively weak, as the combined effect estimation indicates no significant correlation between the two. One potential explanation for this phenomenon is that the connection between emotional exhaustion and UPB is primarily mediated through indirect pathways. For instance, when job insecurity serves as an antecedent variable, emotional exhaustion significantly mediates its relationship with UPB, yet the direct association between job insecurity and UPB is insignificant ([38]). In addition to this, emotional exhaustion, being a multifaceted emotional state, is notably influenced by situational factors. [46] ([46]) discovered that job complexity moderates the relationship between emotional exhaustion and UPB; when job complexity is high, the relationship between UPB and emotional exhaustion is not significant. In contrast, when job complexity is low, emotional exhaustion can positively predict employees’ UPB. Moreover, there are very few studies on UPB and emotional exhaustion, with only five independent samples available for analysis. The limited sample size may exaggerate the issues of publication bias and heterogeneity, making it difficult to obtain a significant and stable overall effect estimate ([17]; [32]).

Two primary behavioral mechanisms have been identified through which the aforementioned negative emotions influence subsequent UPB. These mechanisms are the moral slippery slope and the moral cleansing effect. The moral slippery slope occurs as employees become accustomed to unethical behavior patterns, they are more likely to participate in immoral actions subsequently ([90]; [100]). It is important to note, however, that the occurrence of the moral slippery slope is also influenced by the organizational environment. If leaders condone employees’ involvement in UPB, employees will perceive unethical behavior as acceptable within the organization and are more likely to engage in other unethical behaviors ([16]; [88]).

The moral cleansing effect emphasizes that employees will engage in more altruistic behaviors following UPB to compensate for the harm caused by UPB and thereby restore moral balance ([80]; [98]). The moral cleansing effect actually shares common ground with the moral licensing effect. The moral credits model emphasizes that individuals establish and maintain their moral self-image through past moral behaviors ([53]). Past good deeds are like accumulating moral credits for oneself. Due to the accumulation of moral credit, individuals grant themselves psychological permission to engage in unethical behaviors ([62]). The moral cleansing effect can also be explained by the moral credits model as follows: since UPB undermines the individual’s moral credits, employees need to engage in more moral behaviors to accumulate moral credit and maintain their moral self-image. The fundamental mechanism of the moral cleansing effect lies in alleviating the guilt associated with unethical behavior, emphasizing the driving force of negative feelings.

Finally, there is a stable relationship between the moral emotions generated by UPB and moral disengagement. Regardless of whether individuals experience positive or negative emotions after engaging in UPB, they may resort to moral disengagement to excuse their unethical behavior. [94] ([94]) noted that employees may rationalize UPB through moral disengagement by engaging in cognitive restructuring to release moral self-restraint. The unique nature of UPB enables employees to activate the mechanism of moral disengagement. Although UPB may harm others, its pro-organizational motivation provides employees with a good moral excuse. This justification allows employees to reduce self-reproach and instead regard unethical behavior as an acceptable sacrifice ([21]; [42]).

### 5.3. Theoretical and Practical Implications

On a theoretical level, the present study investigated the emotional response pathways elicited by UPB. It not only disentangled the differential impacts of positive and negative emotions triggered by UPB but also unveiled the potential behavioral mechanisms that may ensue after UPB and the associated emotions, such as the moral cleansing and moral licensing effects. It is worth noting that the meta-analysis indicates that UPB has a stable positive correlation with both negative and positive emotions, which supports the duality paradox of UPB. That is, the concept of UPB itself embodies a duality of pro-organizational and unethical attributes. These two conflicting properties coexist within the construct of UPB, highlighting its paradoxical nature ([45]). Under the influence of pro-organizational and unethical aspects, employees engaging in UPB may feel pride in the pro-organizational outcomes (positive feelings) and shame for their unethical actions (negative feelings). This provides guidance for future researchers to further differentiate the internal components of UPB (pro-organizational and unethical inclinations). Moreover, the inclusion of a large proportion of research by Chinese scholars in the meta-analysis provides a unique perspective for understanding how employees in a collectivist culture may react emotionally and behaviorally to UPB.

In terms of practical significance, the moral slippery slope and moral licensing effects triggered by UPB suggest that people should be vigilant about the moral risks associated with UPB. Considering that moral licensing and moral slippery slope effects often occur in workplace with high workloads or ambiguous ethical norms, organizations not only need to set reasonable performance targets, avoiding excessive workload, but also should place emphasis on the development of an ethical climate, incorporating ethical norms into daily management to enhance employees’ ethical awareness ([88]). Organizations can also strengthen employees’ ability to recognize, analyze, and respond to ethical dilemmas by implementing systematic ethics training programs that convey clear ethical expectations and enhance moral sensitivity ([34]; [68]). Additionally, leaders’ UPB can elicit imitation among organizational members. Therefore, it is necessary to strictly monitor and manage unethical behaviors among the management to prevent the transmission of a self-interested ethical climate ([1]; [42]; [95]). Leadership style also plays a crucial role in preventing unethical behavior. Research has demonstrated that ethical leadership can effectively mitigate employees’ tendencies toward unethical conduct by setting moral examples and establishing ethical norms ([29]; [50]). Therefore, in addition to training the ethical awareness of employees, organizations should actively promote ethical leadership styles among managers. Although the moral cleansing and conscientiousness effects may lead employees to engage in more positive behaviors, the fact remains that employees have still participated in UPB. Tolerance of UPB essentially depletes the organization’s moral capital. Therefore, organizations can provide ethical training and moral decision-making simulations to train employees to identify potential ethical risks in conflicts of interest and to cultivate employees’ positive organizational identification among employees.

### 5.4. Limitations and Implications for Future Research

Although a comprehensive effect analysis of emotions and behavioral mechanisms following UPB has been conducted in this study, several limitations remain. First, this meta-analysis did not include all studies on UPB and emotions. For example, [25] ([25]) found in their research that depression caused by UPB can reduce employee job performance. However, this was not analyzed in the present study. This is because there is only one study on the relationship between UPB and depression, which is insufficient for a comprehensive effect estimation. Nevertheless, we cannot deny the association between UPB and depression, as well as between UPB and other emotions such as anger ([25]; [86]).

However, [69] ([69]) indicated that in addition to emotions such as shame, guilt, and pride, embarrassment; anger, gratitude, and admiration are also common moral emotions, and these moral emotions can influence individuals’ subsequent moral behavior. For example, admiration and gratitude can motivate people to engage in moral actions, while anger and contempt may lead to aggressive behaviors. In research on the relationship between UPB and emotions, scholars have found that hindrance stressors had a positive effect on UPB through the mediation of anxiety and anger ([86]). Insufficient sample sizes limit our ability to assess the strength of the relationship between anger and UPB. However, as a high-arousal and intense negative emotion, anger may drive individuals to act impulsively in response to immediate situations ([9]). According to self-control theory, individuals possess finite self-regulatory resources, and the experience of anger consumes these resources significantly. The depletion of self-control, in turn, promotes unethical behaviors ([94]). Thus, we suspect that anger related to UPB may trigger the moral slippery slope, wherein individuals under anger are more prone to impulsive unethical actions. However, this speculation awaits further empirical validation.

[68] ([68]) discovered that employees who observed others engaging in UPB might experience two types of emotional reactions: admiration and disgust. When individuals admired the UPB actor, they were more likely to offer help to the actor. In contrast, when individuals felt disgusted by the UPB actor, they might report the behavior or avoid interacting with the actor. This suggests that in addition to the emotional-behavioral mechanisms discussed in this paper, UPB may also trigger other emotional-behavioral mechanisms. Although more research is needed to substantiate the behavioral patterns following feelings of disgust or admiration, based on social learning theory ([4]), it can be inferred that when observers admire UPB, they may perceive UPB actors as role models and view UPB as an effective strategy for organizational success, potentially leading to imitation behavior ([42]; [93]). Conversely, disgust toward UPB stems primarily from its unethical nature. According to moral foundations theory, individuals may distance themselves from UPB and its actors due to disgust, as UPB violates two fundamental innate moral principles, “avoid harming others” and “fair reciprocity” ([24]).

The fact that employees experience both disgust and admiration during the UPB process may indicate that the concept itself exhibits duality. This suggests that the pro-organizational and unethical attributes of the concept exert a significant influence on employees’ emotions and behaviors. When employees are more strongly influenced by the pro-organizational attributes of UPB, they are more likely to develop admiration for such behavior. Conversely, when they are more deeply affected by its unethical attributes, they tend to experience disgust. However, traditional UPB scales can only measure employees’ tendencies or behavioral frequency regarding UPB ([52]; [78]), failing to distinguish the extent to which employees are driven by pro-organizational versus unethical motivations during the UPB process. To address this, [45] ([45]) proposed employing the process dissociation paradigm to differentiate the pro-organizational and unethical psychological inclinations underlying UPB. They further developed a new protocol to independently measure these distinct psychological components. Therefore, future research can further explore the associations between UPB and other emotions, and utilize the new measurement tool to examine which component of UPB is attributed to employees’ emotions.

Second, in terms of emotion measurement, most studies have relied on scales to assess employees’ emotional responses. This retrospective self-report method is prone to biases in memory. [59] ([59]) indicated that self-reported emotion scales have difficulty differentiating between transient emotional fluctuations and stable emotional dispositions. Moreover, given that UPB is a behavior that violates general social moral norms, it is more “appropriate” to exhibit guilt in response to such behavior. Therefore, in experimental and survey contexts, participants are easily influenced by social desirability ([57]), resulting in performative emotion reports. To reduce bias, future researchers may consider employing multi-time-point measurements or combining physiological indicators (such as heart rate and skin conductance response) to assist in verifying participants’ emotional changes. Alternatively, [19] ([19]) recommended using the Experience Sampling Method (ESM) to capture emotional changes in real-time, rather than relying solely on retrospective self-reports.

Finally, during the process of literature search and coding, the researchers found that the majority of studies on UPB and emotional reactions came from China, with a significant lack of research and samples from other countries. This phenomenon may be attributed to the following two reasons. From a cultural perspective, China’s strong collectivist culture, characterized by values such as “collective interests first” and “prioritizing the bigger picture,” profoundly shapes the relationship between individuals and organization ([18]; [73]). [84] ([84]) pointed out that Western employment relations tend to follow an impersonal rational economic exchange model, whereas Chinese employment relationships are deeply embedded within a broader sociocultural system. This system emphasizes mutual benefit and reciprocity in interpersonal social relations, which is closely tied to China’s collectivist culture. In such a cultural context characterized by high collectivism and high power distance, loyalty to the organization is often perceived as a virtue ([13]), and this perception further facilitates the rationalization of UPB in Chinese organizations. Second, from a socioeconomic perspective, China is undergoing rapid economic transformation, where enterprises face intense market competition and short-term performance evaluation pressures. To survive or expand, organizations may impose high-performance demands on employees, compelling them to compromise between ethics and task completion, thereby fostering UPB ([12]; [66]). Moreover, during the economic transition period, regulatory systems in certain industries remain under development ([92]). These factors provide space for UPB to survive, allowing it to thrive as an “unwritten rule” within organizations. In summary, collectivist culture provides fertile ground for the emergence of UPB, making it a prominent phenomenon worthy of scholarly investigation. Meanwhile, the rapidly developing market economy generates intense pressures and unique institutional environments, creating abundant cases of UPB and driving research demand. Consequently, a growing number of Chinese scholars have begun focusing on unethical pro-organizational behavior, yielding numerous significant findings.

However, it should be noted that a large number of Chinese samples may indeed affect the ecological validity of the research results, particularly regarding emotions, as the expression of moral emotions varies across different cultures. Some scholars have pointed out that Eastern cultures place a greater emphasis on a culture of shame, in which people experience more shame and embarrassment due to unethical behavior. Meanwhile, Western cultures emphasize a culture of guilt, in which people feel a sense of guilt when engaging in unethical actions ([58]). Accordingly, it can be inferred that the strength of the relationship between UPB and guilt and shame may differ between Eastern and Western cultures. Therefore, the cross-cultural applicability of the meta-analysis results is open to question. Future research may attempt to replicate the aforementioned emotional and behavioral mechanisms in different cultural contexts.

## 6. Conclusions

The present study systematically elucidated the mechanisms through which unethical pro-organizational behavior influences the actor’s emotions and subsequent behaviors. First, UPB can elicit feelings of pride and psychological entitlement in individuals and trigger the moral licensing effect (rationalizing subsequent unethical behavior) and the conscientiousness effect (enhancing organizational identification and promoting positive behaviors). Second, negative emotions such as guilt, shame, and anxiety caused by UPB can drive the moral slippery slope effect (lowering moral standards and engaging in more unethical behaviors) and the moral cleansing effect (repairing self-image through moral behaviors). Finally, as a self-regulatory mechanism, moral disengagement permeates the entire process of UPB, with employees resorting to moral disengagement to resolve the moral conflicts elicited by UPB. The findings of this study reveal the paradoxical nature of UPB, where the coexistence of pro-organizational and unethical attributes within the UPB construct can trigger both positive and negative emotions in individuals. This implies that future scholars should explore the paradoxical nature of UPB and distinguish its inherent duality of conflicting yet coexisting attributes.

## Figures and Tables

**Figure 1 behavsci-15-01266-f001:**
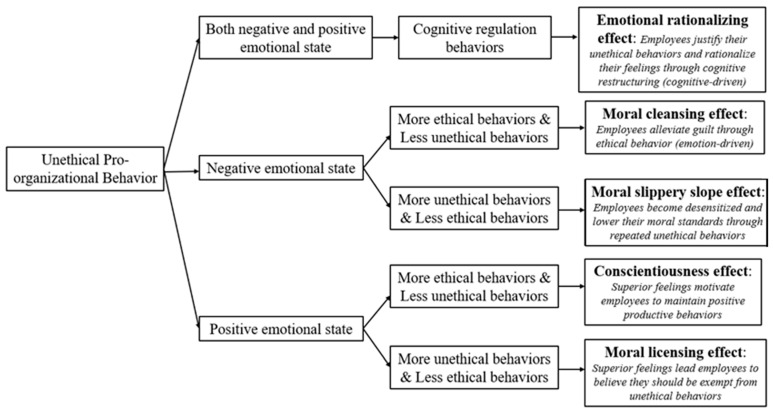
The emotional and behavioral mechanism after UPB.

**Figure 2 behavsci-15-01266-f002:**
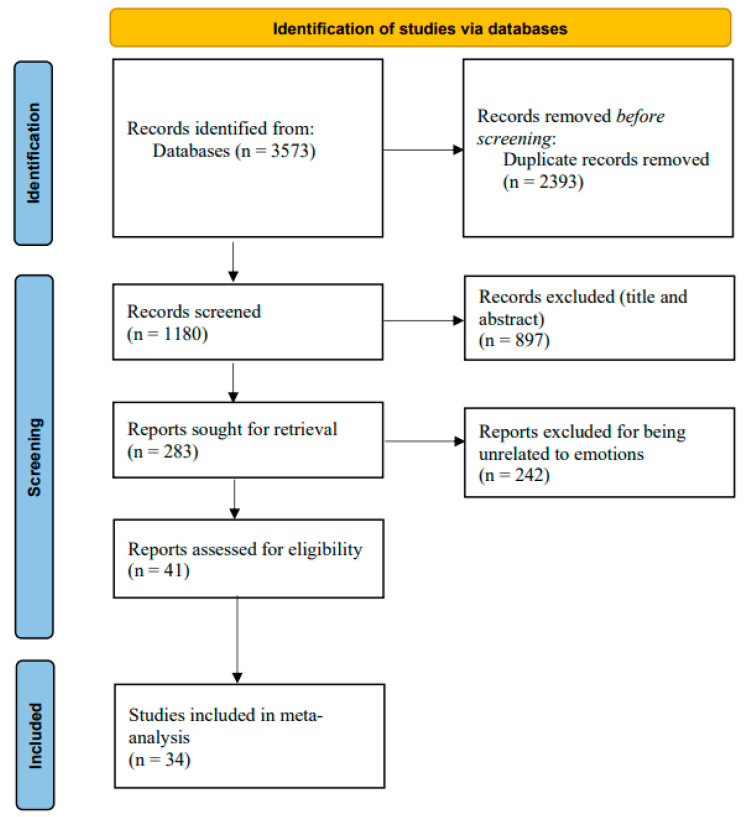
PRISMA flow diagram of study selection process.

**Table 1 behavsci-15-01266-t001:** Heterogeneity test.

Pathway	Type	k	N	Q	df (Q)	T^2^	I^2^
UPB and Emotional responses	pride	7	69,825	78.593 ***	6	0.017	94.747
	guilt	17	72,232	281.669 ***	16	0.062	97.622
	shame	7	68,765	391.841 ***	6	0.041	96.632
	anxiety	8	3308	38.323 ***	7	0.010	80.565
	emotional exhaustion	5	1271	212.452 ***	4	0.198	97.975
	psychological entitlement	22	7531	241.007 ***	21	0.037	92.559
Emotional responses and Behaviors	moral licensing	18	7209	61.128 ***	17	0.007	73.212
	moral slippery slope	13	71,896	320.632 ***	12	0.015	92.784
	moral cleansing effect	10	70,119	206.279 ***	9	0.009	89.146
	conscientiousness effect	12	71,858	570.211 ***	11	0.037	97.114
	emotional rationalizing effect	11	3795	75.413 ***	10	0.018	87.191

k = number of studies included in the analysis; N = total sample size of the studies included in the analysis; *** *p* < 0.001, Q represents the statistic for testing the degree of heterogeneity; df(Q) represents degree of freedom; I^2^ represents the proportion of the heterogeneous part in the total variation of the effect size; OCB refers to organizational citizenship behavior.

**Table 2 behavsci-15-01266-t002:** Publication bias analysis.

Pathway	Type	k	N	Fail-Safe N	*p*-Value
UPB and Emotional responses	pride	7	69,825	517.145	<0.001
	guilt	17	72,232	459.841	<0.001
	shame	7	68,765	390.374	<0.001
	anxiety	8	3308	1220.234	<0.001
	emotional exhaustion	5	1271	4.068	0.652
	psychological entitlement	22	7531	1337.921	<0.001
Emotional responses and Behaviors	moral licensing	18	7209	2048.288	<0.001
	moral slippery slope	13	71,896	817.409	<0.001
	moral cleansing	10	70,119	1537.556	<0.001
	conscientiousness effect	12	71,858	673.728	<0.001
	emotional rationalizing effect	11	3795	1214.929	<0.001

k = number of studies included in the analysis; target significance is 0.05.

**Table 3 behavsci-15-01266-t003:** Overall effect size estimation.

Pathway	Type	Estimate	SE	Z	ES(r)	95% Confidence Interval	
						Lower	Upper
UPB and Emotional responses	pride	0.268	0.053	5.085 ***	0.261	0.163	0.355
	guilt	0.297	0.062	4.795 ***	0.288	0.174	0.395
	shame	0.353	0.080	4.418 ***	0.339	0.194	0.469
	anxiety	0.315	0.040	7.811 ***	0.305	0.232	0.375
	emotional exhaustion	−0.091	0.202	−0.451	−0.091	−0.451	0.295
	psychological entitlement	0.353	0.043	8.179 ***	0.339	0.262	0.411
Emotional responses and Behaviors	moral licensing	0.241	0.024	10.120 ***	0.237	0.192	0.280
	moral slippery slope	0.237	0.037	6.393 ***	0.233	0.163	0.300
	moral cleansing	0.304	0.035	8.768 ***	0.295	0.232	0.356
	conscientiousness effect	0.332	0.057	5.804 ***	0.321	0.217	0.418
	emotional rationalizing effect	0.340	0.044	7.794 ***	0.328	0.249	0.402

*** *p* < 0.001; The pooled effect size ES(r) is transformed using Fisher’s z to r transformation.

## Data Availability

The data detail is available at: https://osf.io/shptd/?view_only=da29f6098d1f4e94b5795e0b7aa58cc5 (accessed on 5 June 2025).

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
