# Peer review of "Emotional and Subsequent Behavioral Responses After Unethical Pro-Organizational Behavior: A Meta-Analysis Based Systematic Review"

_behavsci, 2025, doi:10.3390/bs15091266_

Round 1
Reviewer 1 Report
Comments and Suggestions for Authors
Thank you for the opportunity to read this paper. It tackles and interesting and relevant subject of unethical proorganizational behaviour. Authors nicely introduce the subject and current research gap, with clear theoretical framing. The methodology used is appropriate, and consluions follow main results presented.
Several minor suggestions:
The abstract in its current form could be a bit shortened. I suggest removal of the following „
- In recent years, the emotional responses caused by unethical pro-organiza- 11 tional behavior (UPB) has attracted the attention of scholars. Due to its inherently uneth ical nature, UPB may evoke negative emotions in employees (e.g., guilt), while its perceived pro-organizational intent may simultaneously elicit positive emotions (e.g., pride).“
- state clearly research goals and objectives in the introduction
- While accepted, the line separating "moral cleansing" from "moral compensation" is not always clear.
- Ensure that all constructs are operationalized explicitly and standardize nomenclature.
- Authors acknowledge that most included studies are from China still implications of this need to be more thoroughly discussed.
- Expand the discussion on how underrepresented emotions may alter or extend the emotional-behavioral pathways.
- Make text more fluent
Author Response
Thank you for taking time to read our manuscript and providing us with valuable and constructive suggestions that will help us improve it.
Please see the attachment for our response.

Reviewer 2 Report
Comments and Suggestions for Authors
Dear authors
The study's topic is highly relevant and timely, addressing a growing interest in understanding the paradoxical effects of UPB not only on organizations but also on the employees who enact such behaviors. The attempt to categorize post-UPB responses into discrete emotional-behavioral mechanisms is commendable and provides useful insight for both scholars and practitioners.
In terms of consistency between the title, abstract, and keywords, the manuscript is largely coherent. The title captures the central theme by emphasizing the emotional and behavioral responses following UPB and references the use of a meta-analytic systematic review. However, a minor but important language issue arises here: the term “Systematical Review” is grammatically awkward and should be revised to “Systematic Review” for clarity and correctness. The abstract effectively summarizes the theoretical framing, methodology, and findings, and aligns well with the title. The keywords are appropriate and aligned with the manuscript content, although some refinement could improve discoverability. For example, replacing the broad term "emotional-behavioral mechanism" with more specific concepts such as “moral licensing,” “self-regulation,” or “psychological entitlement.”
The manuscript is well-structured, logically transitioning from conceptual background to methodological execution and results. The literature review demonstrates substantial effort to organize the existing body of research by emotion types (positive and negative) and corresponding behaviors, culminating in the proposal of five emotional-behavioral mechanisms: moral cleansing, moral licensing, moral disengagement, moral slippery slope, and conscientiousness effects. This classification offers a valuable framework to map the dual valence of emotional responses to UPB. However, the discussion occasionally becomes overly descriptive and repetitive, especially in the treatment of pride, guilt, and psychological entitlement, and would benefit from further synthesis to improve readability and conceptual sharpness. Additionally, while the authors distinguish among the five effects, the underlying mechanisms sometimes appear overlapping. A clearer conceptual differentiation, perhaps using a comparative table or matrix, would enhance clarity.
From a theoretical standpoint, the authors draw primarily from moral disengagement theory, affective appraisal theory, and conservation of resources (COR) theory. These are appropriate and well-justified. However, the theoretical contribution could be strengthened by incorporating additional perspectives such as Social Cognitive Theory of Morality or Self-Determination Theory (SDT), which could better account for the complex self-regulatory processes involved. Moreover, the paper presents a unidirectional emotion-to-behavior sequence without considering feedback loops or reciprocal influences between behavioral engagement and moral emotion. This limits the comprehensiveness of the emotional-behavioral model.
A notable omission in the theoretical framing is the literature on the paradoxical consequences of positive organizational behaviors. I would strongly encourage the authors to consider integrating or at least citing the following relevant and recent study:
Sulistiawan, J., Sumarsono, J. J. P., Lin, P. K., & Dwikesumasari, P. R. (2024). Navigating the Thin Line: Can Acts of Good Lead to Defiance at Work? Exploring the Intricacies of OCB and CWB Dynamics. Public Integrity, 1–18. https://doi.org/10.1080/10999922.2024.2380401
This paper explores how seemingly prosocial behaviors (e.g., Organizational Citizenship Behavior) may paradoxically foster deviant outcomes like Counterproductive Work Behavior (CWB), often mediated by psychological entitlement or perceived moral license. The insights from that study would complement and deepen the current manuscript’s framework by bridging the literature between ethical and unethical prosocial acts and their emotional aftermath.
Methodologically, the manuscript follows the PRISMA guidelines for systematic review and includes a substantial sample of 83,810 participants across 34 studies and 49 independent samples. The use of Fisher’s Z transformation, heterogeneity statistics (Q and I²), and the Paule-Mandel method is statistically sound and appropriately justified given the observed heterogeneity. Quality assessments using the AHRQ tool are conducted, although it is worth noting that this tool is primarily designed for healthcare and may not be the best fit for behavioral studies. One major limitation is the overrepresentation of studies conducted in China, which raises questions about cross-cultural generalizability. While the authors do acknowledge this in the limitations section, they stop short of testing culture as a moderator, which could have added significant depth to the analysis.
Moreover, the study does not include moderator or subgroup analyses despite observing high heterogeneity across emotional-behavioral pathways. For example, variables such as organizational ethical climate, type of industry, or individual moral identity could significantly moderate the relationships between UPB, emotions, and behavior. Their exclusion limits the explanatory power of the findings. Additionally, although the study discusses emotional exhaustion as a potential outcome, it reports a non-significant correlation between UPB and emotional exhaustion, which contradicts several theoretical expectations. This warrants further exploration, potentially as an indirect path rather than a direct outcome.
Finally, the manuscript’s discussion and conclusion effectively summarize the findings and offer actionable theoretical and practical implications. Particularly strong is the discussion on the dual consequences of UPB—how the same behavior can generate moral disengagement for some and moral cleansing for others. However, the discussion could be enhanced by offering clearer guidelines on how organizations might prevent the moral licensing and slippery slope effects, especially in settings with strong performance pressures or ambiguous ethical norms.
In conclusion, this manuscript offers a solid foundation for understanding the emotional and behavioral consequences of UPB through a well-executed meta-analytic lens. However, to strengthen its theoretical and empirical contribution, I recommend revisions in the following areas: (1) revising the title for clarity and accuracy; (2) refining the emotional-behavioral mechanism distinctions; (3) integrating or citing recent paradox literature such as Sulistiawan et al. (2024); and (4) addressing heterogeneity more rigorously through moderator analysis or clearer acknowledgment of cultural bias. With these improvements, the paper would make a meaningful contribution to the literature on ethics, emotions, and employee behavior.
Regards.
Author Response
Thank you for taking the time to read our manuscript and providing us with valuable and constructive suggestions that help us improve it.
Please see the attachment for our responses.

Reviewer 3 Report
Comments and Suggestions for Authors
Thank you for the opportunity to review the journal article: Emotional and Subsequent Behavioral Responses After Unethical Pro-organizational Behavior: A Meta-analysis Based Systematical Review. In times of technical developments and automation, the aspect of ethics increases in relevance.
Based on the nature of the paper as a systematic literature review, there is a diverse engagement with the literature and a description of the relevant aspects in relation to the topic. Nevertheless, I would appreciate a more critical engagement with the literature, synthesising the different ideas further. It often leaves me with the question 'so what?' when reading the article.
The theme of description is present in the abstract as well, which would benefit from being more succinct but elaborating further on the findings, theories, and implications for the academic conversation.
The introduction does not guide the reader about what to expect. What has been the overall research question or phenomenon? What is your line of argument that you try to prove with the literature review?
The use of Chinese databases should have been disclosed earlier, as I would argue it has a significant impact on the findings due to cultural context.
The theoretical and practical implications provide an opportunity for improvement by anchoring the theoretical perspective in relational analysis with the journal article. The practical implications could be more concrete, highlighting who would have the responsibility to implement them.
The conclusion leaves me with the same question - so what? What is your argument and contribution to the discourse?
Smaller suggestions:
I would not start the abstract with the phrase 'in recent years' as it is dating the article and not contributing to any clarification.
Could a translation of the keywords for the Chinese databases be included to make it accessible for a wider audience?
What does NO. stand for?
Be consistent with the reference guidelines, which potentially do not permit numbers under ten as they need to be written out.
Could you include a summary overview of findings beyond the statistics? Your description is dense and dry, so helping the reader with a visualisation could be beneficial.
Please consider an alphabetical order of the keywords and abbreviations
Author Response

(The authors gave the same response as above.)

Reviewer 4 Report
Comments and Suggestions for Authors
I was curious to read the paper. It is very easy to read and to understand and is clear in focus and regarding the structure. I have several points which can improve the paper.
- I suggest to add a section in which unethical pro-socual behavior is defined against ethical pro-organizations behavior and pro-social rule breaking. There are several similar constructs in organizational research and I was wondering how these terms differ or share characterists.
- The introduction is quite comprehensive and interesting to read-nevertheless I was wondering why authors did not end with hypothesis which guide the meta-analysis? Figure 1 is summarizing the theoretical introduction in a descriptive way- why do authors do not add specific hypothesis they test subsequently?
-
I suggest to give concrete examples of what kind of unethical pro-organizational behavior was investigated in the studies that were included in the meta-analysis. The description of the studies included is somewhat superficial.
- Additionally, I would liket to get more information on the measures used in measuring UPB and emotional responses- where all measures the same? Was there any effect based on the questionnaires or measurement instruments used?
- I would also like to receive more information about the severity of unethical behavior. I would expect that there is a range of unethical pro-organizational behavior- from "lying to a customer" to severe fraud.
-
Please add also the kind of organizations, and services or products provided and the persons-status in the organisation (employee or supervisior/manager). There should be information on the persons included in the N of the studies of interest.
-
In the discussion I was missing practical implication regarding Compliance Management, HRM and supervisior and managerial behavior. What can e.g. recruitment, training, performance management contribute to reduce unethical behavior?
- The section on the limitation on cultural effects are interessing- while I was reading the study I was also wondering why most of the studies have authors (names) which made me think of an asian context - and the authors write about a majority of studies coming from China. What is a possible explanation?
In the current form the paper adresses an intersting topic, but much more information is needed to understand the studies and the organizations involved in the meta-analysis.
Author Response

(The authors gave the same response as above.)

Round 2
Reviewer 2 Report
Comments and Suggestions for Authors
Dear authors,
Thank you for revising your manuscript and addressing my comments and suggestions.
The revised manuscript demonstrates a commendable effort by the authors to address the concerns raised in the initial review. The revisions are thoughtful, substantial, and directly responsive to the key points previously identified, particularly regarding conceptual clarity, theoretical robustness, and integration of relevant literature.
1. Title, Abstract, and Keywords
The revision of the title from “Systematical Review” to the grammatically correct “Systematic Review” is appreciated. The updated keywords now include more specific constructs such as “moral licensing effect,” “moral cleansing effect,” “moral slippery slope,” and “moral self-regulation,” enhancing both clarity and discoverability. These adjustments resolve the earlier concern regarding awkward phrasing and overly broad descriptors.
2. Conceptual Clarity of Emotional-Behavioral Mechanisms
The manuscript now presents a more nuanced and synthesized discussion of the five proposed emotional-behavioral effects following UPB. The revised Literature Review section elaborates on the core distinctions between moral cleansing and moral compensation (licensing), and the authors have added comparative conceptual framing that clarifies overlaps and differentiates constructs more explicitly. The update to Figure 1 is particularly helpful in delineating the distinct psychological mechanisms. This revision greatly improves the readability and theoretical sharpness of the manuscript.
3. Theoretical Expansion and Citation Integration
The authors incorporated Social Cognitive Theory to strengthen the explanatory power of the proposed self-regulatory mechanisms, particularly in relation to moral disengagement and imitation behavior. While Self-Determination Theory (SDT) was not incorporated due to scope alignment concerns, the rationale is clearly explained and acceptable given the study’s aims.
Importantly, the authors acknowledged the suggestion to integrate the paradoxical outcomes of prosocial behavior and have appropriately cited Sulistiawan et al. (2024) in the discussion section. This addition effectively enriches the theoretical framework by connecting OCB-induced psychological entitlement with the moral licensing pathway, which strengthens the manuscript’s positioning within current ethical behavior discourse.
4. Methodological Rigor and Limitations
The meta-analytic methodology continues to follow PRISMA guidelines and is sound. The authors provided a more detailed justification for the use of the AHRQ tool despite its healthcare origins, noting its suitability given the predominance of cross-sectional designs in the reviewed studies. Although moderator and subgroup analyses (e.g., cultural moderators or ethical climate) were not added due to sample limitations, the authors have offered a compelling justification supported by specific numeric breakdowns of available data points.
Furthermore, the revised manuscript acknowledges the non-significant findings for emotional exhaustion and expands the discussion with plausible theoretical and methodological explanations, including the potential role of indirect pathways, job complexity, and sample size limitations. This transparency reflects strong scientific integrity.
5. Practical Implications
The practical implications section has been expanded to include actionable organizational strategies to prevent the slippery slope and moral licensing effects, especially under high workload or ambiguous ethical climates. The authors now discuss the role of ethical leadership, training, and the ethical climate more concretely, providing a useful translation of findings into actionable recommendations.
The authors have addressed all major points from the initial review, demonstrating a willingness to improve both conceptual depth and methodological transparency. The manuscript now offers a clearer and more robust contribution to the literature on UPB, self-regulation, and moral emotions. Therefore, based on my second review, I recommend this manuscript for acceptance. Congratulations to all of the authors for their hard work.
Author Response
Thank you very much for your kind and encouraging feedback on our revised manuscript. We are deeply grateful for the time and effort you have invested in reviewing our work and for your constructive comments, which have significantly improved the quality of our paper.
We are delighted to hear that you found our revisions thoughtful and substantial, and that the manuscript now meets the standards for acceptance. Your insightful suggestions throughout the review process have been invaluable in enhancing the conceptual clarity, theoretical robustness, and overall coherence of our work.
Thank you again for your supportive comments and for recommending our manuscript for acceptance. We greatly appreciate the opportunity to contribute to the literature and are thrilled that our efforts have been positively received.
Reviewer 3 Report
Comments and Suggestions for Authors
It is great to see that you reflected on the peer reviews and included various, if not all, suggestions, which shows in the increased quality of the paper. After revision, often new areas of improvement arise, so don't be discouraged by the following comments.
The abstract still has room to advocate for your research clearly. In my opinion, it still provides too much background information, leaving not enough space to highlight your insights from the research project.
Figure 1 has moral disengagement floating around without any connections with the rest of the model.
Reflecting on the Chinese context again, I wonder if it would be more rigorous to strip out non-Chinese studies to be exact about the findings and theoretical and practical implications.
Otherwise, as mentioned before, great improvement.
Author Response
Thank you for taking the time to read our manuscript and providing us with valuable and constructive suggestions that help us improve it.
Please see the attachment for authors' responses.
